# Regenerated *Antheraea pernyi* Silk Fibroin/Poly(*N*-isopropylacrylamide) Thermosensitive Composite Hydrogel with Improved Mechanical Strength

**DOI:** 10.3390/polym11020302

**Published:** 2019-02-11

**Authors:** Boxiang Wang, Song Zhang, Yifan Wang, Bo Si, Dehong Cheng, Li Liu, Yanhua Lu

**Affiliations:** 1School of Materials Science and Engineering, Shanghai University, Shanghai 200444, China; bxwang0411@163.com (B.W.); zsong0928@shu.edu.com (S.Z.); xiaolv526@sina.com (Y.W.); 2Key Laboratory of Functional Textile Materials, Liaoning Province, Eastern Liaoning University, Dandong 118003, China; chengdehongldxy1@163.com; 3School of Chemical Engineering, Eastern Liaoning University, Eastern Liaoning University, Dandong 118003, China; sbsw004403@163.com

**Keywords:** *Antheraea pernyi*, silk fibroin, hydrogel, thermosensitive, mechanical properties

## Abstract

At present, *Antheraea pernyi* silk fibroin (ASF) has attracted research efforts to investigate it as a raw material for fabrication of biomedical devices because of its superior cytocompatibility. Nevertheless, native ASF is not easily processed into a hydrogel without any crosslinking agent, and a single hydrogel shows poor mechanical properties. In this paper, a series of ASF/poly (N-isopropylacrylamide) (PNIPAAm) composite hydrogels with different ASF contents were manufactured by a simple in situ polymerization method without any crosslinking agent. Meanwhile, the structures, morphologies and thermal properties of composite hydrogels were investigated by XRD, FTIR, SEM, DSC and TGA, respectively. The results indicate that the secondary structure of silk in the composite hydrogel can be controlled by changing the ASF content and the thermal stability of composite hydrogels is enhanced with an increase in crystalline structure. The composite hydrogels showed similar lower critical solution temperatures (LCST) at about 32 °C, which matched well with the LCST of PNIPAAm. Finally, the obtained thermosensitive composite hydrogels exhibited enhanced mechanical properties, which can be tuned by varying the content of ASF. This strategy to prepare an ASF-based responsive composite hydrogel with enhanced mechanical properties represents a valuable route for developing the fields of ASF, and, furthermore, their attractive applications can meet the needs of different biomaterial fields.

## 1. Introduction

Hydrogels have been investigated as suitable materials for tissue engineering, cell culture, drug carriers, and artificial cartilage because of their high water content and tissue-like elastic properties. They can mimic the natural extracellular matrix in terms of bioactivity [1]. Their unique degradation, flexible mechanical properties and tunable compositions provide the opportunity to achieve various biological functions that are beneficial for cell engineering. Simultaneously, the mechanical strength and surface topography of hydrogels can be controlled in order to mimic tissue cultural environments for anchorage-dependent cell adhesion [2]. 

Natural structural proteins display critical structural and bioactive properties, including long-range ordered molecular secondary structures and high primary amino acid sequences that have the broader application of functional protein-based composite biomaterials [3]. It is well known that silk fiber is a natural structural protein with excellent characteristics, such as high tensile strength, Young’s modulus, toughness and extensibility that is superior to synthetic fiber [3]. At present, silk fibroin (SF) comes to the fore in all realms except the dress field because of its excellent and unique properties, such as nontoxicity, biocompatibility and biodegradability. SF is extensively used in enzyme immobilization materials [4,5,6], food packaging materials [7], hydrogel materials [8,9], tissue engineering [10,11,12], and drug delivery carriers [13,14,15]. However, there is still the challenge of how to control the mechanical properties of SF-based biomaterials, especially silk fibroin hydrogels.

Silks can be classified as mulberry and non-mulberry, which are produced by domesticated *Bombyx mori* and wild silkworm species, respectively [16]. *Antheraea pernyi* is a wild non-mulberry silkworm species belonging to the *Saturniidae* family, which is commonly known as Chinese temperate (oak) tussah [17]. There is a significant difference in primary amino acid composition of *Antheraea pernyi* silk fibroin (ASF) and *Bombyx mori* silk fibroin (BSF). The amino acid sequences of silk fibroin largely vary from species to species, which results in a wide range of properties [18]. ASF is composed of alanine (43.07%), glycine (27.27%), serine (11.26%), tyrosine (5.26%) and aspartic acid (4.47%) [19,20]. ASF contains inherent arginyl-glycyl-aspartic (RGD) tripeptide sequences, distinguishing it from other silk fibroins, which are binding sites of cell integrin receptors [21,22]. These RGD sequences mediate the interactions between mammalian cells and extracellular matrices [23]. The RGD sequence in ASF provides stronger adhesion of cells compared to BSF [24,25,26]. Therefore, ASF has attracted research efforts to investigate it as a raw material for fabrication of biomedical devices, due to its superior cytocompatibility. In addition, it has been approved three kinds of conformation in SF (ASF and BSF), which are α-helix, β-sheet and random coil [27]. The crystalline structures of both the SFs are classified as β structures (β-sheet, β-turn). There is a strong resistance of *A. pernyi* fiber to chemicals in the presence of abundant β-sheets. Regenerated ASF is usually prepared by dissolving *A. pernyi* fibers at high temperature in a calcium nitrate solution or concentrated aqueous lithium thiocyanate solution. The aqueous solution of ASF is unstable and sensitive to environment temperatures, which change the α-helix and random coil to a β-sheet. The gelation transformation of ASF solution happens rapidly [28], but the material prepared with ASF is not easily processed into different geometrical structures and a lack of practical applications has been because of these poor mechanical properties. Therefore, the biomedical potential of ASF is not fully explored due to this problem. At present, biomaterials based on BSF have been widely researched and applied in biomedical fields [29,30]. However, the research on using ASF in biomaterials is only at a preliminary stage compared with BSF because of the poor mechanical properties limiting the application of ASF as a biomaterial [31]. In order to overcome this drawback, chemical crosslinking agents like formaldehyde, glutaraldehyde, and epoxy can be employed to prepare hydrogels and films with excellent mechanical properties. However, these crosslinking agents may cause residual toxicity. At present, mixing low or non-toxicity polymers is an effective approach to generate ASF-based biomaterials that have good biocompatibility, special functions and flexible mechanical properties. Blending proteins is a technological approach to generate protein-based biomaterials with a more complete set of specific properties [3]. 

Thermo-responsive materials have earned considerable interest due to their value for both scientific research and practical pharmaceutical applications [32,33,34,35]. Poly (N-isopropylacrylamide) (PNIPAAm) is the most intensively and widely investigated stimuli-responsive polymer. It reveals a volume phase-transition in response to even slight temperature changes. The PNIPAAm polymer exhibits a temperature dependent phase transition in aqueous solution at 32 °C, the lower critical solution temperature (LCST) [36], and has the ability to switch between a hydrophobic and hydrophilic surface at different physiologically acceptable temperatures [37,38]. Because the LCST of PNIPAAm is near body temperature, it has the sharpest phase transition of all thermosensitive N-alkylacrylamide polymers, and has often been utilized in tissue engineering [39,40], thermally modulated drugs [41,42] and gene delivery systems [43,44]. Previously, thermo-responsive cell culture materials enabled the control of cell adhesion and detachment by changing the temperature. In recent years, the most successful applications of a thermo-responsive cell culture material are dishes [45], hydrogels [46] and membranes [47] prepared by PNIPAAm. In spite of these outstanding features, pristine PNIPAAm hydrogels have been notorious for their poor mechanical properties, hindering their applications in various fields over the past decades [48,49]. Therefore, synthesis of PNIPAAm composites has been conducted to address these limitations. Many researchers focus on the preparation of PNIPAAm-based functional composites with inorganic particle or other polymers to enlarge the potential applications of PNIPAAm. However, PNIPAAm composite hydrogels with natural protein are insufficient in research.

In this work, in order to overcome the shortcomings of ASF and PNIPAAm hydrogels, we propose a new and facile method to fabricate ASF/PNIPAAm composite hydrogels by a simple in situ polymerization method. There is no crosslinking agent in this composite system. With the crystalline structure of ASF, the composite hydrogels showed the desired excellent mechanical properties. Moreover, the hydrogels showed temperature sensitive properties by differential scanning calorimetry (DSC). The temperature dependences of the equilibrium swelling ratio of the hydrogels were characterized. Meanwhile, the morphological, crystallization change and thermal decomposition of the resultant product were investigated by means of scanning electron microscopy (SEM), wide angle X-ray diffraction (WAXD), and thermogravimetric analysis (TGA). In particular, the secondary structure changes of the composite hydrogels were studied by peak differentiating and imitating of Fourier transform infrared spectroscopy (FTIR). By the combination of ASF and PNIPAAm, we aimed to make composite hydrogels that possess thermo-responsiveness, without any crosslinking agent, and enhanced mechanical strength.

## 2. Materials and Methods

### 2.1. Materials

Raw silk cocoons of *Antheraea pernyi* (Liaoning Tussah Silk Institute Co., Ltd., Dandong, China) were used for preparation of ASF. *N*-isopropylacrylamide (NIPAAm) (Aladdin Bio-Chem Technology Co., Ltd., Shanghai, China, 98%) was purified by dissolution and recrystallization in hexane. Ammonium persulfate (APS), *N,N,N′,N′*-tetramethylethylenediamine (TEMED), sodium carbonate (Na_2_CO_3_), calcium nitrate (Ca(NO_3_)_2_) were purchased from Sinopharm Chemical Reagent Co., Ltd., Shanghai, China.

### 2.2. Preparation of Regenerated ASF

Regenerated ASF was extracted following the procedure as per earlier established literature [8,16,19]. Briefly, raw cocoons of *A. pernyi* were degummed by boiling three times in 0.25% (w/v) Na_2_CO_3_ solution at 100 ± 2 °C or 30 min to extract the traces of sericin, rinsed thoroughly with distilled water, and dried in an oven. After drying, the degummed fibers were dissolved in molten Ca(NO_3_)_2_ at 105 ± 2 °C for 4 h. The rough yellow regenerated ASF solution was then filtered. The mixed solution was dialyzed against deionized water for 3 days with an 8–14 kDa molecular weight cut off dialysis tube to remove salts. The solution was finally centrifuged at 8000 rpm for 10 min to remove impurities. The pure regenerated ASF solution was frozen at −40 °C for 12 h, followed by freeze-drying for 48 h to obtain regenerated ASF (Figure 1).

### 2.3. Synthesis of ASF/PNIPAAm Composite Hydrogels

Firstly, 75mg/mL NIPAAm monomer aqueous solution was prepared. Subsequently, a certain amount of NIPAAm aqueous solution and the initiator APS were mixed with the as-prepared regenerated ASF, followed by stirring for 1 h to prepare the hydrogels. Then, TEMED as an activator was additionally dissolved in an as-prepared mixed solution. The solution was injected into a glass mold and dry nitrogen gas was bubbled into the solution for 30 min. In the end, the glass mold was quickly covered with a Teflon stopper. The solution mixture was polymerized by the free radical reaction after incubating at room temperature for 12 h. After polymerization, the hydrogel samples were dialyzed against deionized water for 24 h with an 8–14 kDa molecular weight cut off dialysis tube to remove unreacted chemicals.

In this work, the total solid concentration was 50 mg/mL and the initiator was 0.2% (w/w) of the monomer content. The prepared hydrogels were noted as AxNy, where the subscripts x and y represent the mass ratio of ASF and NIPAAm, respectively. The formulations of the composite hydrogels are listed in Table 1.

### 2.4. Wide-Angle X-ray Diffraction (WAXD)

X-ray powder diffraction was conducted to analyze the crystalline state of the ASF and composite hydrogels by using an X-ray diffractometer (D/max-2550, Rigaku, Japan). Cu-Kα radiation was used for the X-ray source at 40 kV and 30 mA. All scans were performed from 5–45° (2θ) at a speed of 2°/min.

### 2.5. Fourier Transform Infrared Spectroscopy (FT-IR)

The structure of composite hydrogels was analyzed by FT-IR (Nicolet IS10 FT-IR spectrometer, Thermo Fisher Scientific, USA). The pure sample solids were prepared in KBr (Sigma-Aldrich) pellets for FT-IR spectra analysis. All infrared spectra were recorded in the range of 4000–500 cm^−1^ using an accumulation of 32 scans with a resolution of 4 cm^−1^. The second-derivative and cure-fitting of the infrared spectra covering the amideIregion (1700–1600 cm^−1^) were performed using peak-fit software to identify the secondary structure of hydrogels [50].

### 2.6. Thermogravimetric Analysis (TGA)

The composite hydrogels of different ASF content were determined from thermogravimetric analysis (TGA), using a TGA system (Q500, TA instruments, USA). The hydrogel samples were heated to 550 °C with a step increase of 5 °C/min under an inert nitrogen atmosphere.

### 2.7. Differential Scanning Calorimetry (DSC)

The LCSTs of the swollen ASF/PNIPAAm composite hydrogel samples were determined using a differential scanning calorimeter (Q2000, TA instruments, USA) under nitrogen at a flow rate of 20 mL/min. The weight of the swollen sample was kept at about 10 mg. All samples were performed at 0–50 °C at a heating rate of 2 °C/min.

### 2.8. Swelling Ratio Measurement

The swelling ratio of hydrogel samples was gravimetrically monitored by immersing the dried hydrogel samples (*W_d_*) in excessive PBS (pH = 7.4) in the range from 15 °C to 50 °C. The mass of swollen hydrogels (*W_s_*) was measured after removing the surface water. The experimental results were calculated from an average of three samples. The equilibrium swelling ratio (ESR) was defined as the weight of absorbed water per weight of the dried hydrogel. The ESR was calculated as follows:(1)$$\mathrm{Equilibrium}\;\mathrm{Swelling}\;\mathrm{Ratio}=\frac{W_{s}-W_{d}}{W_{d}}$$

### 2.9. Mechanical Properties

The compressive property of ASF/PNIPAAm composite hydrogels was determined by using a texture analyzer (TMS-PRO, FTC, USA) at 25 °C and 65% ± 5% R.H. Cylindrical hydrogel samples were prepared with flat and parallel surfaces, cut as uniform strips with an initial diameter and height of 10 ± 0.5 mm and 8 ± 0.5 mm, respectively. The test speed was set at 5 mm/min, the trigger force was 0.5 N and the deformation was 50% under compression mode. Three samples for each group were tested (n ≥ 3).

### 2.10. Morphology of ASF/PNIPAAm Composite Hydrogels

The composite hydrogel samples were freeze-dried and quenched in liquid nitrogen, followed by sputtering with a thin layer of gold. Then the composite hydrogels were observed using a scanning electron microscope (JSM-IT100, JEOL, Japan) at 20 kV.

## 3. Results and Disscussion

### 3.1. Preparation of ASF/PNIPAAm Composite Hydrogels

PNIPAAm hydrogels have been extensively studied for drug carriers, tissue engineering and gene delivery [23,24,25,26]. However, the linear PNIPAAm can only take the shape of a hydrogel in the presence of a crosslinking agent. ASF hydrogels can be prepared from aqueous ASF solution by formatting the physical crosslinking structure, which involves the α-helix and random coil transforming into a β-sheet. The (-Ala-)_n_ polypeptide sequences made ASF much more hydrophobic and resulted in the gelation transition of ASF being much more rapid than the other silk fibroin. The gelation behavior of ASF solution and the thermosensitive behavior of linear PNIPAAm are shown in Figure 2. It can be seen (Figure 2a) that the yellow transparent ASF solution (30 mg/mL) becomes white gel after it had been placed for a while. However, the ASF gel still flowed and crushed after gentle shaking and inversion of the sample bottle. Although the gelation transition of ASF solution happened rapidly, it could not form a stable shape hydrogel by itself. The colorless transparent linear PNIPAAm solution became white when the temperature exceeded the LCST, seen in Figure 2b. Although the linear PNIPAAm solution was thermosensitive, it could still flow after inversion of the sample bottle. It is difficult for the linear PNIPAAm to turn into a hydrogel without any crosslinking agent. This result indicated that both ASF and PNIPAAm cannot form hydrogels with geometric shapes by a single component.

ASF/PNIPAAm hydrogels were fabricated by mixing solutions of ASF with NIPAAm monomer. The free radical polymerization of the hydrogels was carried out in a glass mold under initiator APS and catalysis of initiator TEMED at room temperature for 12 h. The linear PNIPAAm reacted with the ASF macromolecular peptide chain entanglement, and ultimately formed an excellent shape hydrogel. In this research, stabilized hydrogel can take shape by mixing a low concentration of ASF solution (5 mg/mL) and in-situ polymerized PNIPAAm without any crosslinking agent (Figure 3).

### 3.2. Structure Characteristics

Figure 4 shows the WAXD date for (a) *A. pernyi* silk fiber (ASF) and (b) composite hydrogels of different ASF content. The ASF (Figure 4a) shows a typical X-ray diffractogram of the silkI structure, having broad diffraction peaks at 2θ values of 11.5°, 14.4°, 22.2°, 25.4° and 29.4°. The *A.p.* silk fiber and composite hydrogels exhibited the typical crystalline structure at 2θ values of 16.9°, 20.3° and 24.3°, corresponding to the silkII crystalline structure. But the A4N6 sample (with 0.4 ASF fractions) also showed obvious silkI crystal structure, having the same diffraction peaks as the ASF. Compared with other hydrogel samples, there were new peaks at 19.8°, 20.4° and enhanced strength of the silkI peaks at 14.4°, 25.4° and 29.4°, corresponding to the silkI crystalline structure. The results indicated that both silkI and silkII formed with increased ASF content. ASF fractions in composite hydrogels have a significant influence on the formation of the silkI crystalline structure.

Structural changes in the ASF/PNIPAAm composite hydrogels after different ASF contents were confirmed by FTIR (Figure 5). The infrared spectral region within 1700–1500 cm^−1^ is assigned to absorption by the peptide backbones of amideI (1700–1600 cm^−1^) and amideII (1600–1500 cm^−1^), which have been commonly used for the analysis of different secondary structures of silk fibroin. In the present study, the amideI band for ASF shows one strong peak at 1658 cm^−1^, corresponding to an α-helix. However, the amideI band of composite hydrogels shows one strong peak at 1639 cm^−1^, with a shoulder peak at 1658 cm^−1^, corresponding to a β-sheet and α-helix, respectively. The analysis of the major peaks observed in the spectra of composite hydrogels showed the presence of intense peaks at 1531 cm^−1^, corresponding to the N–H flexural vibration and C–N stretching vibration, and 1247 cm^−1^, corresponding to the C–N–H of PNIPAAm. With the addition of ASF, the spectra of blend hydrogel appeared at 966 cm^−1^ and 620 cm^−1^ which correspond to the poly-alanine fragments [51]. It did not show new chemical bonds were formed in FTIR. This indicates that the ASF was not chemically bound to PNIPAAm but was physically interpenetrated within the PNIPAAm hydrogel.

As reported, the shape of the amideI band was used to determine the protein secondary structure. Located at 1700–1600 cm^−1^ was the β-sheet (1610–1642 cm^−1^, 1680–1700 cm^−1^), random coil (1642–1650 cm^−1^), α-helix (1650–1660 cm^−1^) and β-turn (1660–1680 cm^−1^) [52]. The secondary structure of ASF/PNIPAAm hydrogels with different ASF weight fractions was further characterized by cure-fitting of the spectra (Figure 6). The fractional small peaks were assigned to each secondary component and the percentage of the secondary structural element was determined by quantifying the assigned peak area (Table 2). The composite hydrogels (even with 0.1 ASF fractions (A1N9)) showed significant β-sheet fraction (approximately 61%), which was similar to the *A.pernyi* silk fiber. This implied that the amide bands of composite hydrogels were governed mainly by the silk component. A4N6 had the highest content of β-turn (30.64%) and the fraction of crystalline structure (β-turn and β-sheet 91.8%) was the highest among all hydrogel samples. In this study, silkI crystalline structure was formed at an appropriate ratio to allow sufficient space for ASF chain self-assembly. The silkI structure was considered a necessary intermediate for the pre-organization or pre-alignment of silk fibroin molecules in the natural silk spinning process [53]. The FTIR results were consistent with the WAXD results, which confirmed that the secondary structure of silk can be controlled by changing the content of ASF. It was revealed that ASF self-assembling into the silkI structure in composite aqueous systems was an inherent ability and some factors such as ionic strength, temperature, concentration and pH can affect the self-assembly process. Therefore, various aspects of performance could be affected by the silkI crystalline structure in ASF based composite materials.

### 3.3. Thermodynamic Properties of ASF/PNIPAAm Hydrogels

The thermal stability of composite hydrogels with different ASF content was characterized by thermogravimetric analysis. Figure 7 shows the TG and DTG curves for ASF and composite hydrogels. The ASF sample showed obvious decomposition peaks at 303.7 °C and 351.2 °C. The aggregation structure of regenerated silk fibroin became a disorder incompact state because of the peptides of ASF were rearranged by solvent dissolving. The peak 1 of ASF was the side-chain group degradation. The ASF/PNIPAAm hydrogels with different ASF content showed two decomposition peaks at around 350–410 °C. The peak 2 can be attributed to the main-chain degradation of ASF. It can be seen that peak 2 of the samples was gradually increased to range between A1N9 (355.5 °C) and A4N6 (359.6 °C) (Table 3). The peptide chain of ASF showed a more orderly arrangement due to the intermolecular hydrogen bonding between ASF and PNIPAAm during the process of gelation. The better thermal stability of composite hydrogels was due to the cross-linked network formation and stronger crystalline structure. The peak 3 can be attributed to the degradation of the PNIPAAm chain. The decomposition temperature was improved with increased PNIPAAm content. The TGA results confirmed that a stronger interaction and increased crystalline structure between ASF and PNIPAm were the reasons for good thermal stability.

### 3.4. Thermosensitivity of ASF/PNIPAAm Hydrogels

The LCSTs of the series of the hydrogel samples were examined by DSC (Figure 8). The onset temperature of endotherm was referred to as the LCST. Obviously, all the samples showed a similar LCST at about 32 °C, which matched well with the LCST of the PNIPAAm hydrogel. The composite hydrogels with different ASF content showed that there was little impact on the LCST of PNIPAAm. The results indicated that the networks between ASF and PNIPAAm are chemically identical. Although, no reaction occurred between the peptide chain of ASF and PNIPAAm. However, it was shown that with increased ASF content, the LCST of composite hydrogels slightly declines. It is well known that the LCST is the point where the hydrophobic interaction of the isopropyl group of PNIPAAm outweighs the hydrophilic nature of the amide group in the pendant group, forcing water out of the hydrogel [54]. In present study, ASF consisted of a large amount of poly-alanine, which is super hydrophobic. Therefore, hydrophobicity was reinforced by the abundant amount of poly-alanine in the hydrogel network.

The temperature dependence of ESR is shown in Figure 9. The swelling ratio dates here show that all the hydrogels have a similar classical thermosensitive profile. The ESRs of all the hydrogels decreased dramatically towards their LCST and had the sharpest decrease around 32 °C, where the phase separation occurred. When temperature was under the LCST, the hydrogels showed different levels of ESR with different mass ratios of ASF to NIPAAm. This can also be illustrated by the enhancement of hydrophobicity that plenty of poly-alanine caused. Therefore, the ESR was reduced by increasing the ASF content. The LCST from the ESR observation was coincided to the thermal date of the DSC curves.

### 3.5. Compression Mechanical Properties of ASF/PNIPAAm Hydrogels

Many applications of PNIPAAm hydrogels are in aqueous medium, in which they swell to a very high degree. This results in low density of the polymer chains which makes the gels extremely poor in physical strength [55]. However, the mechanical profiles required in biological applications are even more diverse and difficult to achieve. The mechanical properties of the scaffold for cell growth should match that of the host tissue [56]. In order to investigate the influence of ASF and NIPAAm content on the mechanical properties of the composite hydrogels, compression tests were conducted. In the stress–strain curve (Figure 10a), it is shown that with an increase in strain, each hydrogel reveals the no fracture phenomenon in the strain of 50%. The compressive stress was gradually increased to 50% strain. Enhanced mechanical properties were found with a higher weight fraction of ASF (Figure 10b). With the content of ASF, the elasticity modulus (Figure 10b) gradually improved from 80 ± 23 kPa (A1N9) to 606 ± 93 kPa (A4N6). There was approximately an 8-fold increase in elasticity modulus with a 0.3 ASF weight fraction (A4N6) compared to A1N9. This behavior can be ascribed to the enhanced crystalline structure (β structure) within the composite hydrogels, as well as the contribution of hydrogen bonding between ASF peptide chains and polymer chains. However, further increasing the ASF content induced a decline in elasticity modulus (A5N5, 467 ± 61kPa). With more ASF, superfluous peptide chains may aggregate with a strong hydrophobicity interaction. Such aggregation, if any, may deteriorate the mechanical properties. However, this mechanical property can be still achieved in order to mimic the range of elasticity of different tissues [57,58].

### 3.6. Morphology of ASF/PNIPAAm Hydrogels

According to the SEM images of ASF/PNIPAAm composite hydrogels, the obtained hydrogels with different ASF contents show porous structure after lyophilization (Figure 11 (A)–(E)). The average pore size of composite hydrogels ranged from 200 μm (A1N9) to 60 μm (A5N5), which was dependent on the content of ASF. The more ASF, the more pore size will be uniform. The increasing content of ASF may cause the 3D network structure to be more uniform and average pore size to decrease. Therefore, the ESR of hydrogels decreased with enhanced network structure. This result was consistent with the observation of the ESR date. Furthermore, Figure 11 (a)–(e) illustrates the inner structure of pores. The nanowires in the hydrogel framework can be seen. These nanowires are the trace during the gelation of ASF. In the process of gelation, nanowires slowly formed the network structure by means of weak forces (included hydrogen bonds, ionic bonds, water-mediated hydrogen bonds, hydrophobic and van der Waals interactions, etc.) [59] without any intervention. This process of self-assembly can affect the transition of crystalline structure, even finally impacting on the properties of composite hydrogels. Therefore, further research in this field must continue.

## 4. Conclusions

The ASF/PNIPAAm composite hydrogels were prepared by mixing ASF and NIPAAm with a simple in situ polymerization method. Thermosensitive and robust composite hydrogels could take shape without any crosslinking agent. The LCSTs of the prepared composite hydrogels were detected at about 32 °C, which was similar to PNIPAAm. The content of ASF in composite hydrogels had a significant influence on the formation of the silkI crystalline structure. These obtained composite hydrogels demonstrated a crystalline structure and enhanced mechanical properties, which can be regulated by changing the mixing ratio of ASF to NIPAAm within the composite hydrogels. The secondary structure of silk in composite hydrogels, which can be controlled by changing the content of ASF, was the reason for their high mechanical strength. Furthermore, these hydrogels had a favorable thermal stability property and porous structure, which may present a valuable route for developing the fields of ASF and their attractive applications for meeting the needs of different biomaterial fields. These tough and responsive hydrogels may have potential applications for scaffold materials, artificial cartilage, cell culture, cell migration and smart soft actuators.

## Figures and Tables

**Figure 1 polymers-11-00302-f001:**
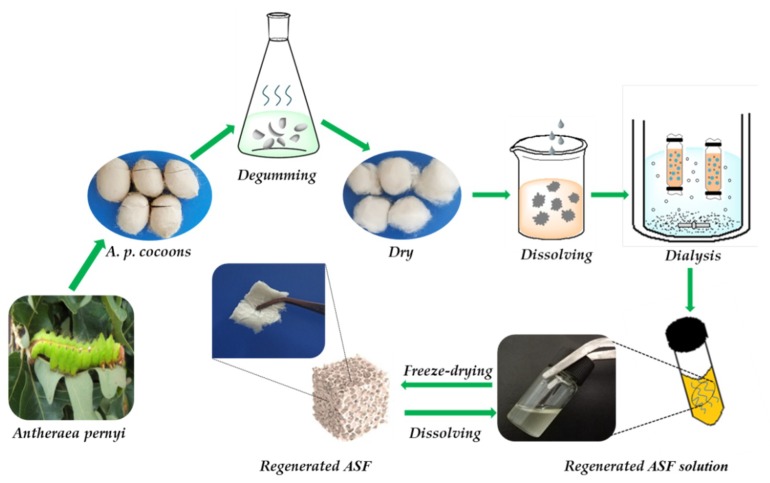
Preparation of regenerated ASF.

**Figure 2 polymers-11-00302-f002:**
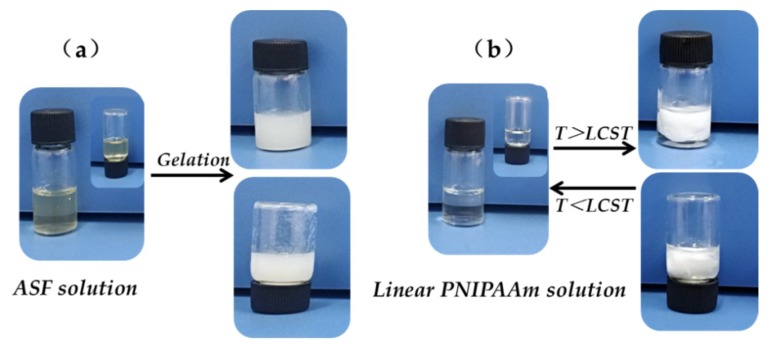
(**a**) Gelation behavior of ASF solution; (**b**) thermosensitive behavior of linear PNIPAAm solution.

**Figure 3 polymers-11-00302-f003:**
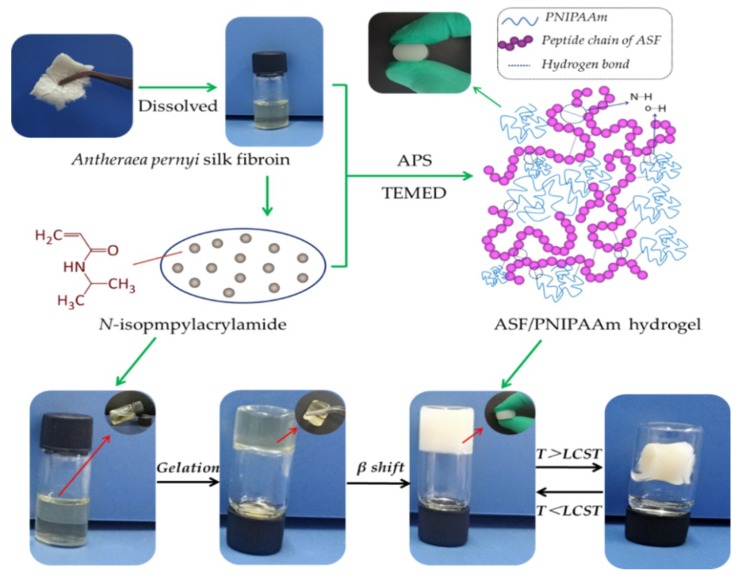
Preparation process of ASF/PNIPAAm composite hydrogel.

**Figure 4 polymers-11-00302-f004:**
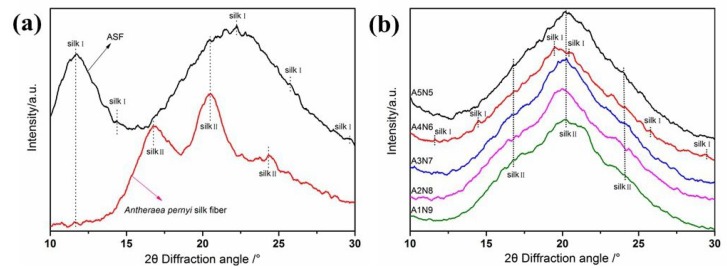
XRD graph of (**a**) *A.p**ernyi* silk fiber and ASF; (**b**) composite hydrogels of different ASF content.

**Figure 5 polymers-11-00302-f005:**
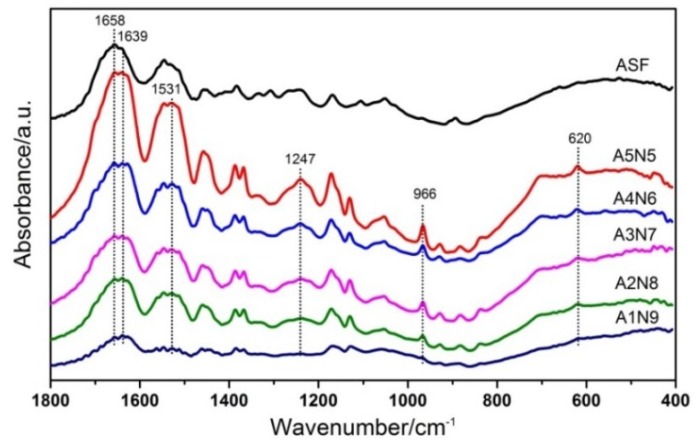
FTIR absorbance spectra of different composite hydrogels.

**Figure 6 polymers-11-00302-f006:**
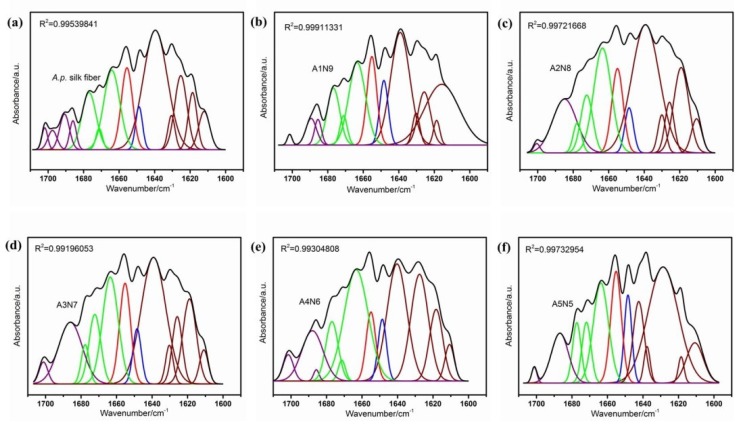
FTIR absorbance spectra and curve-fitting results in the amideI band for (**a**) *A.p**ernyi* silk fiber (**b**–**f**) composite hydrogels of different ASF content.

**Figure 7 polymers-11-00302-f007:**
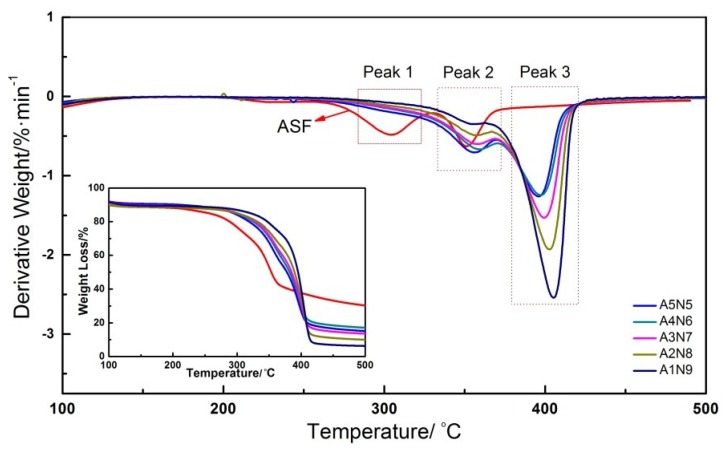
TGA curves for composite hydrogels prepared by different ASF content.

**Figure 8 polymers-11-00302-f008:**
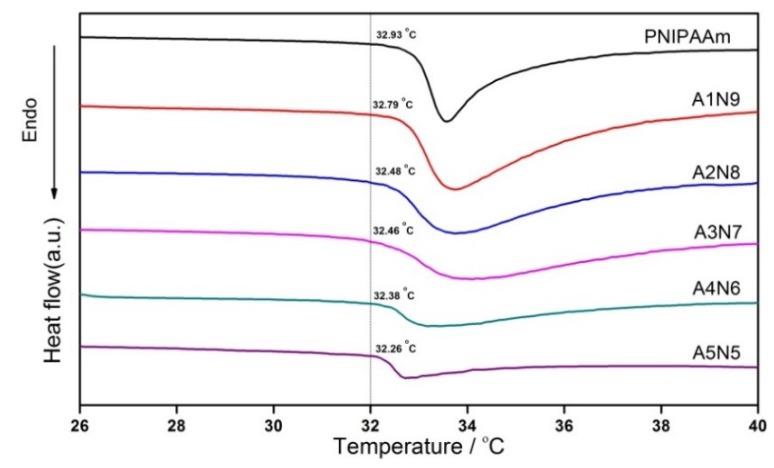
DSC curves for composite hydrogels prepared by different ASF content.

**Figure 9 polymers-11-00302-f009:**
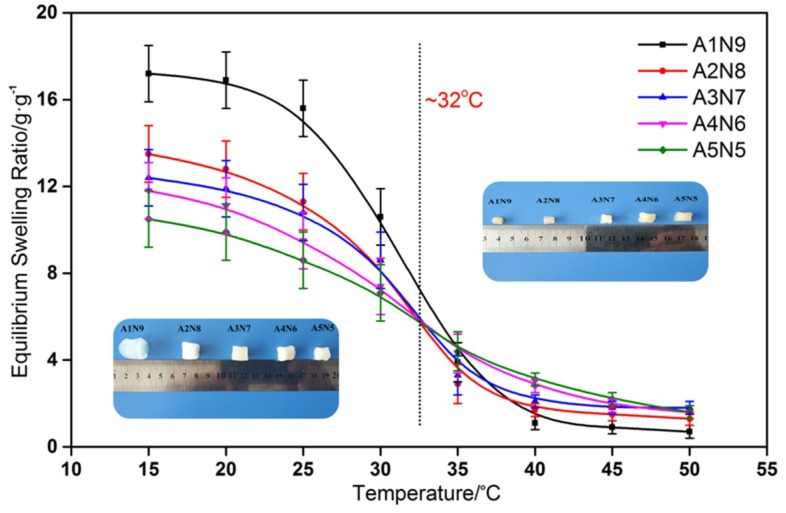
ESR for composite hydrogels in the temperature range from 15 °C to 50 °C.

**Figure 10 polymers-11-00302-f010:**
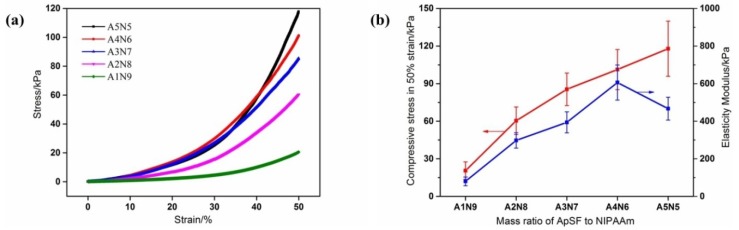
Compression mechanical properties: (**a**) stress–strain curves of composite hydrogels with different ASF content; (**b**) average compressive stress in 50% strain and elasticity modulus with different mixing ratios.

**Figure 11 polymers-11-00302-f011:**
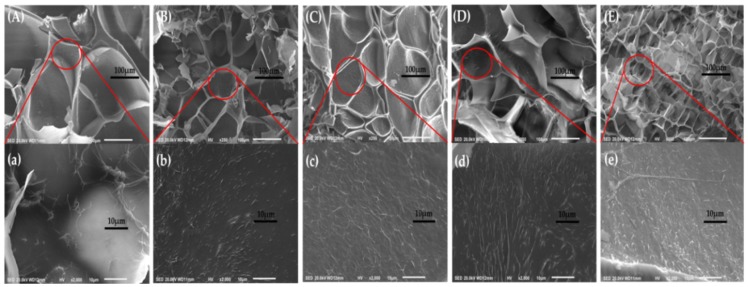
SEM images of composite hydrogels with different ASF content: (**A**, **a**) A1N9, (**B**, **b**) A2N8, (**C**, **c**) A3N7, (**D**, **d**) A4N6, (**E**, **e**) A5N5.

**Table 1 polymers-11-00302-t001:** The formulations of ASF/PNIPAAm composite hydrogels.

Code	ASF:NIPAAm (Mass Ratio)	ASF (mg)	NIPAAm (mL)	APS (mg)	5%TEMED (µL)
A1N9	1:9	52.5	6.3	0.95	19
A2N8	2:8	105	5.6	0.84	16.8
A3N7	3:7	157.5	4.9	0.74	14.8
A4N6	4:6	210	4.2	0.63	12.6
A5N5	5:5	262.5	3.5	0.53	10.6

**Table 2 polymers-11-00302-t002:** Secondary structure of ASF in the composite hydrogels with different mixing ratios.

Assignment	*A.p.* fiber (%)	A1N9 (%)	A2N8 (%)	A3N7 (%)	A4N6 (%)	A5N5 (%)
β-sheet (silkII)	61.61	60.99	61.29	61.81	61.16	61.24
β-turn (silkI)	24.13	24.16	24.75	24.41	30.64	27.32
α-helix (silkI)	10.01	9.08	9.94	9.37	4.73	7.43
Random coil (silkI)	4.25	5.77	4.02	4.41	3.47	4.01

**Table 3 polymers-11-00302-t003:** TGA date of composite hydrogels.

Assignment	ASF	A1N9	A2N8	A3N7	A4N6	A5N5
Peak 1 (°C)	303.7	-	-	-	-	-
Peak 2 (°C)	351.2	355.5	357.4	357.5	359.6	355.7
Peak 3 (°C)	-	405.3	402.7	399.5	399.2	396.1

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
