# Peer review of "Regenerated Antheraea pernyi Silk Fibroin/Poly(N-isopropylacrylamide) Thermosensitive Composite Hydrogel with Improved Mechanical Strength"

_polymers, 2019, doi:10.3390/polym11020302_

Round 1

Reviewer 1 Report

In this manuscript titled “Regenerated Antheraea pernyi silk fibroin/poly (N-isopropylacrylamide) thermosensitive composite hydrogel with high mechanical strength. The authors prepared ASF/PNIPAAm composite hydrogels by mixing ASF and NIPAAm without crossing agent, to obtain a valuable biomaterial with favorable thermal stability and high mechanical strength. The manuscript is well organized  and delivered with logic and convincing results.

1.     In Figure 4b, ASF fraction in composite hydrogel has significant influence on the silk I crystalline structure. How do you explain A4N6 sample showed more obvious silk I crystal structure than the other sample, especially A5N5 ?

2.     By which kind of physical interaction should ASF and NIPPAm form hydrogel without chemical reaction ?

Author Response

Response 1: Thanks for your question.

According to the literature [(1)Qiang L. Xiao H. Water-Insoluble Silk Films with Silk I Structure. ACTA BIOMATER. 2009, 6, 1380-1387. (2)Jinfa M. Baoqi Z. Silk I structure formation through silk fibroin self-assembly. Journal of Applied Polymer Science 2012, 125, 2148-2154], silk fibroin gelation is essentially a self-assembly process. By controlling the drying rate, the silkstructure of the films can be prepared directly from silk fibroin aqueous solution. Since with very slow drying silk fibroin has sufficient time to self-assemble into stable structures.

In this paper, most of the water quickly collected the PNIPAAm around when the ASF content was low. Since PNIPAAm is super hydrophilic at room temperature. This may causes the self-assembly process of ASF happens rapidly and stable β-sheet crystalline structuresilkⅡ) is formed directly. But with the increase of the mass ratio of ASF and NIPAAm, the speed of ASF self-assembly gradually slows down. By this time, the ability of PNIPAAm binding water was reduced with the decreased of the PNIPAAm content. There is enough time and space in the system for ASF self-assemble. And β-turn crystalline structuresilkⅠ)can be formed. This result was similar to the reports in the literature.

A4N6 sample can show more obvious silkcrystalline structure than other sample, especially A5N5. This may be that the interaction between ASF and PNIPAAm just reaches an equilibrium point. In this condition, ASF self-assembly process is more stable. When the ASF content continues to increase, the ASF may be excessive in the A5N5 sample. The interaction between ASF itself gradually increased in addition of the interaction between ASF and PNIPAAm. A part of ASF directly formed β-sheet crystalline structuresilkⅡ)under this condition. Therefore, it is not especially obvious of silkcrystalline structure in A5N5 sample compared with A4N6 sample. But the silkcrystalline structure still can be seen in the XRD curves of A5N5.

Response 2: Thank you for your question. In the process of in-situ polymerization, stable interpenetrating network structure hydrogel was formed by ASF and PNIPAAm. In this process, hydrogen bond interactions between ASF and PNIPAAm were the major physical interaction to form hydrogel.

Reviewer 2 Report

The manuscript was written well and the author has justified his objective with proper experimental design.

I have few comments to be addressed

 1. In figure 2 its not looking like hydrogel, it seems only the colour of the solution has been changed, describe in details in the caption for better understanding.                                      2. Material prepared in this study was formed by chemical or physical interaction?. 

3.explain why the intensity of the peak decrease in FTIR? 
4. in the sem image it is not looking like the size of the pores decreases, while looking at the image B, C, D it shows the pore size decreases? describe in details

Author Response

Response 1: Yes. Thanks for your question. In figure 2 both (a) and (b) are not hydrogel. The detail describe has been revised in the original paper. It has been replenished in line 201-207. (Although the gelation transition of ASF solution happened rapidly but it cannot form stable shape hydrogel itself. The colorless transparent linear PNIPAAm solution becomes white when the temperature exceeds the LCST in Figure 2(b). Although the linear PNIPAAm solution was thermosensitive but still can flow after inversion of sample bottle. It is difficult that the linear PNIPAAm turn into stable shape hydrogel without any crosslinking agent. This result indicated that both ASF and PNIPAAm cannot form hydrogels with geometric shapes by single component.)

Response 2: It is a good question. In the process of in-situ polymerization, stable interpenetrating network structure hydrogel was formed by ASF and PNIPAAm. In this process, hydrogen bond interactions between ASF and PNIPAAm were the major physical interaction to form hydrogel.

Response 3: Thanks for your question. The infrared spectral region within 1700-1500 cm-1 is assigned to absorption by the peptide backbones of amide(1700-40000px-1) and amide(1600-1500 cm-1), which have been commonly used for the analysis of different secondary structures of silk fibroin.

Since ASF content is different in each sample so the intensity of the peak decreases in FTIR. The ordinate of FT-IR presents the relative absorption intensity. Hence, the relative intensity of each peak is decreased in single drawing. Each curve will present a strong peak when it exists independently. It is similar to the literature [Yifan W. Li L. Conductive graphene oxide hydrogels reduced and bridged by L-cysteine to support cell adhesion and growth. J. Mater. Chem. B, 2017, 5, 511].

Response 4: Thanks for your question. It is a good suggestion. It is my fault to describe the SEM images. I feel very sorry for my carelessness. It was found that not all pore size was decreased with the increase of ASF content. With the increase of ASF content, the average pore size tends to decrease at the image B, C and D. When the content of ASF was low, the self-assembly process of ASF may be limited to a space of the hydrogel. Hence, the pore size was not uniform. The increasing content of ASF may cause the 3D network structure more uniform and average pore size was decreased. The more ASF, the more pores size will be uniform. It has been replenished in line 341-344.

Reviewer 3 Report

In this manuscript, the authors presented the fabrication of ASF and PNIPAAm hydrogels and systematically characterized the properties of hydrogel. In general, this manuscript is well presented. The English language should be significantly improved.

Comments

The title should be changed. Based on the mechanical properties results, the mechanical strength is not high.

For mechanical property test, please specify the test condition and parameters

Some potential applications of the composite should be discussed.

Author Response

Response 1: Thank you for your guidance. The title (high mechanical strength) has been changed (improved mechanical strength).

Response 2: Thank you for your suggestion. The test speed was set at 5mm/min, the trigger force was 0.5N and the deformation was 50% under compression mode. It has been replenished in line 183-184 (2.9 Mechanical properties).

Response 3: Thank you for your suggestion. Some potential applications have been discussed in the conclusion. It has been replenished in line 368-369. (These tough and responsive hydrogels may have potential applications for scaffold materials, artificial cartilage, cell culture, cell migration and smart soft actuators.)

Round 2

Reviewer 3 Report

All the issues have been addressed